# Is Internet Pornography Causing Sexual Dysfunctions? A Review with Clinical Reports

**DOI:** 10.3390/bs6030017

**Published:** 2016-08-05

**Authors:** Brian Y. Park, Gary Wilson, Jonathan Berger, Matthew Christman, Bryn Reina, Frank Bishop, Warren P. Klam, Andrew P. Doan

**Affiliations:** 1Flight Surgeon, Fleet Logistics Support Squadron 40, Norfolk, VA 34800 Bob Wilson Drive, San Diego, CA 92592, USA; brian.y.park4.mil@mail.mil; 2The Reward Foundation, 5 Rose Street, Edinburgh EH2 2PR, Scotland, UK; GWilson@rewardfoundation.org; 3Department of Urology, Naval Medical Center San Diego, 34800 Bob Wilson Drive, San Diego, CA 92592, USA; Jonathan.Berger@med.navy.mil (J.B.); matthew.christman@med.navy.mil (M.C.); 4Department of Mental Health, Naval Medical Center San Diego, 34800 Bob Wilson Drive, San Diego, CA 92592, USA; bryn.reina@med.navy.mil (B.R.); warren.p.klam.civ@mail.mil (W.P.K.); 5Department of Ophthalmology, Naval Medical Center San Diego, 34800 Bob Wilson Drive, San Diego, CA 92592, USA; frank.bishop@med.navy.mil

**Keywords:** erectile dysfunction, low sexual desire, low sexual satisfaction, delayed ejaculation, pornography, Internet pornography, sexually explicit material, PIED

## Abstract

Traditional factors that once explained men’s sexual difficulties appear insufficient to account for the sharp rise in erectile dysfunction, delayed ejaculation, decreased sexual satisfaction, and diminished libido during partnered sex in men under 40. This review (1) considers data from multiple domains, e.g., clinical, biological (addiction/urology), psychological (sexual conditioning), sociological; and (2) presents a series of clinical reports, all with the aim of proposing a possible direction for future research of this phenomenon. Alterations to the brain's motivational system are explored as a possible etiology underlying pornography-related sexual dysfunctions. This review also considers evidence that Internet pornography’s unique properties (limitless novelty, potential for easy escalation to more extreme material, video format, etc.) may be potent enough to condition sexual arousal to aspects of Internet pornography use that do not readily transition to real-life partners, such that sex with desired partners may not register as meeting expectations and arousal declines. Clinical reports suggest that terminating Internet pornography use is sometimes sufficient to reverse negative effects, underscoring the need for extensive investigation using methodologies that have subjects remove the variable of Internet pornography use. In the interim, a simple diagnostic protocol for assessing patients with porn-induced sexual dysfunction is put forth.

## 1. Introduction

### 1.1. Trends in Sexual Dysfunction—Unanswered Questions

Up until the last decade, rates of ED were low in sexually active men under 40, and did not begin to rise steeply until thereafter [1,2]. A 1999 major cross-sectional study reported erectile dysfunction in 5%, and low sexual desire in 5% of sexually active men, ages 18 to 59 [3], and a 2002 meta-analysis of erectile-dysfunction studies reported consistent rates of 2% in men under 40 (except for the preceding study) [2]. These data were gathered before Internet “porn tube sites” enabled wide access to sexually explicit videos with no download required. The first of these “tube sites” appeared in September 2006 [4].

In contrast, recent studies on ED and low sexual desire document a sharp increase in prevalence of such dysfunctions in men under 40. One clear demonstration of this phenomenon relates to ED, and compares very large samples, all of which were assessed using the same (yes/no) question about ED as part of the Global Study of Sexual Attitudes and Behavior (GSSAB). In 2001–2002, it was administered to 13,618 sexually active men in 29 countries [5]. A decade later, in 2011, the same (yes/no) question from the GSSAB was administered to 2737 sexually active men in Croatia, Norway and Portugal [6]. The first group, in 2001–2002, were aged 40–80. The second group, in 2011, were 40 and under. Based on the findings of historical studies cited earlier, older men would be expected to have far higher ED rates than the negligible rates of younger men [2,7]. However, in just a decade, things changed radically. The 2001–2002 rates for older men 40–80 were about 13% in Europe [5]. By 2011, ED rates in young Europeans, 18–40, ranged from 14%–28% [6].

In the last few years, research using a variety of assessment instruments has revealed further evidence of an unprecedented increase in sexual difficulties among young men. In 2012, Swiss researchers found ED rates of 30% in a cross-section of Swiss men aged 18–24 using the International Index of Erectile Function (IIEF-5) [8]. A 2013 Italian study reported one in four patients seeking help for new onset ED were younger than 40, with rates of severe ED nearly 10% higher than in men over 40 [9]. A 2014 study on Canadian adolescents reported that 53.5% of males aged 16–21 had symptoms indicative of a sexual problem [10]. Erectile dysfunction was the most common (26%), followed by low sexual desire (24%), and problems with orgasm (11%). The results took the authors by surprise, “It is unclear why we found such high rates overall, but especially the high rates among both male and female participants rather than female participants alone, as is common in the adult literature” [10] (p.638). A 2016 study by this same group assessed sexual problems in adolescents (16–21 years) in five waves over a two-year period. For males, persistent problems (in at least one wave) were low sexual satisfaction (47.9%), low desire (46.2%), and problems in erectile function (45.3%). The researchers noted that over time rates of sexual problems declined for females, but not for males [11]. A 2014 study of new diagnoses of ED in active duty servicemen reported that rates had more than doubled between 2004 and 2013 [12]. Rates of psychogenic ED increased more than organic ED, while rates of unclassified ED remained relatively stable [12]. A 2014 cross-sectional study of active duty, relatively healthy, male military personnel aged 21–40 employing the five-item IIEF-5 found an overall ED rate of 33.2% [13], with rates as high as 15.7% in individuals without posttraumatic stress disorder [14]. The researchers also noted that sexual dysfunctions are subject to underreporting biases related to stigmatization [14], and that only 1.64% of those with ED had sought prescriptions for phosphodiesterase-5 inhibitors through the military [13]. A second analysis of the military cross-sectional data revealed that the increased sexual functioning problems were associated with “sexual anxiety” and “male genital self-image” [14]. A 2015 “Brief Communication” reported ED rates as high as 31% in sexually active men and low sexual desire rates as high as 37% [6]. Finally, another 2015 study on men (mean age approximately 36), reported that ED accompanied by a low desire for partnered sex is now a common observation in clinical practice among men seeking help for their excessive sexual behavior, who frequently “use pornography and masturbate” [15].

Traditionally, ED has been seen as an age-dependent problem [2], and studies investigating ED risk factors in men under 40 have often failed to identify the factors commonly associated with ED in older men, such as smoking, alcoholism, obesity, sedentary life, diabetes, hypertension, cardiovascular disease, and hyperlipidemia [16]. ED is usually classified as either psychogenic or organic. Psychogenic ED has been related to psychological factors (e.g., depression, stress, generalized anxiety, or performance anxiety) while organic ED has been attributed to physical conditions (e.g., neurological, hormonal, anatomical, or pharmacologic side effects) [17]. For men under 40 the most common diagnosis is psychogenic ED, and researchers estimate that only 15%–20% of cases are organic in origin [18].

However, none of the familiar correlative factors suggested for psychogenic ED seem adequate to account for a rapid many-fold increase in youthful sexual difficulties. For example, some researchers hypothesize that rising youthful sexual problems must be the result of unhealthy lifestyles, such as obesity, substance abuse and smoking (factors historically correlated with organic ED). Yet these lifestyle risks have not changed proportionately, or have decreased, in the last 20 years: Obesity rates in U.S. men aged 20–40 increased only 4% between 1999 and 2008 [19]; rates of illicit drug use among US citizens aged 12 or older have been relatively stable over the last 15 years [20]; and smoking rates for US adults declined from 25% in 1993 to 19% in 2011 [21]. Other authors propose psychological factors. Yet, how likely is it that anxiety and depression account for the sharp rise in youthful sexual difficulties given the complex relationship between sexual desire and depression and anxiety? Some depressed and anxious patients report less desire for sex while others report increased sexual desire [22,23,24,25]. Not only is the relationship between depression and ED likely bidirectional and co-occurring, it may also be the consequence of sexual dysfunction, particularly in young men [26]. While it is difficult to quantify rates of other psychological factors hypothesized to account for the sharp rise in youthful sexual difficulties, such as stress, distressed relationships, and insufficient sex education, how reasonable is it to presume that these factors are (1) not bidirectional and (2) have mushroomed at rates sufficient to explain a rapid multi-fold increase in youthful sexual difficulties, such as low sexual desire, difficulty orgasming, and ED?

### 1.2. Is Internet Pornography Use a Factor In Today's Sexual Dysfunctions?

Kinsey Institute researchers were among the first to report pornography-induced erectile-dysfunction (PIED) and pornography-induced abnormally low libido, in 2007 [27]. Half of subjects recruited from bars and bathhouses, where video pornography was “omnipresent”, were unable to achieve erections in the lab in response to video porn. In talking to the subjects, researchers discovered that high exposure to pornography videos apparently resulted in lower responsivity and an increased need for more extreme, specialized or “kinky” material to become aroused. The researchers actually redesigned their study to include more varied clips and permit some self-selection. A quarter of the participants’ genitals still did not respond normally [27].

Since then, evidence has mounted that Internet pornography may be a factor in the rapid surge in rates of sexual dysfunction. Nearly six out of 10 of 3962 visitors seeking help on the prominent “MedHelp.org ED Forum”, who mentioned their ages, were younger than 25. In that analysis of eight years of posts and comments, among words commonly linked with the mental aspect of ED (non-organic ED), “porn” appeared most frequently by far [28]. A 2015 study on high school seniors found that Internet pornography use frequency correlated with low sexual desire [29]. Of those who consumed Internet pornography more than once a week, 16% reported low sexual desire, compared with 0% in non-consumers (and 6% for those who consumed less than once a week). Another 2015 study of men (average age 41.5) seeking treatment for hypersexuality, who masturbated (“typically with very frequent pornography use”) seven or more hours per week, found that 71% had sexual dysfunctions, with 33% reporting difficulty orgasming [30]. Anxiety about sexual performance may impel further reliance on pornography as a sexual outlet. In a 2014 functional magnetic resonance imaging (fMRI) study, 11 of the 19 compulsive Internet pornography users (average age 25), whose brains were scanned for evidence of addiction, reported that as a result of excessive use of Internet pornography they had “experienced diminished libido or erectile function specifically in physical relationships with women (although not in relationship to the sexually explicit material)” [31]. Clinicians have also described pornography-related sexual dysfunctions, including PIED. For example, in his book *The New Naked*, urology professor Harry Fisch reported that excessive Internet pornography use impairs sexual performance in his patients [32], and psychiatry professor Norman Doidge reported in his book *The Brain That Changes Itself* that removal of Internet pornography use reversed impotence and sexual arousal problems in his patients [33]. In 2014, Bronner and Ben-Zion reported that a compulsive Internet pornography user whose tastes had escalated to extreme hardcore pornography sought help for low sexual desire during partnered sex. Eight months after stopping all exposure to pornography the patient reported experiencing successful orgasm and ejaculation, and succeeded in enjoying good sexual relations [34]. To date, no other researchers have asked men with sexual difficulties to remove the variable of Internet pornography use in order to investigate whether it is contributing to their sexual difficulties.

While such intervention studies would be the most illuminating, our review of the literature finds a number of studies that have correlated pornography use with arousal, attraction, and sexual performance problems [27,31,35,36,37,38,39,40,41,42,43], including difficulty orgasming, diminished libido or erectile function [27,30,31,35,43,44], negative effects on partnered sex [37], decreased enjoyment of sexual intimacy [37,41,45], less sexual and relationship satisfaction [38,39,40,43,44,45,46,47],a preference for using Internet pornography to achieve and maintain arousal over having sex with a partner [42], and greater brain activation in response to pornography in those reporting less desire for sex with partners [48]. Again, Internet pornography use frequency correlated with low sexual desire in high school seniors [29]. Two 2016 studies deserve detailed consideration here. The first study claimed to be the first nationally-representative study on married couples to assess the effects of pornography use with longitudinal data. It reported that frequent pornography consumption at Wave 1 (2006) was strongly and negatively related with participants’ marital quality and satisfaction with their sex life at Wave 2 (2012). The marriages most negatively affected were those of men who were viewing pornography at the highest frequencies (once a day or more). Assessing multiple variables, the frequency of pornography use in 2006 was the second strongest predictor of poor marital quality in 2012 [47]. The second study claimed to be the only study to directly investigate the relationships between sexual dysfunctions in men and problematic involvement in OSAs (online sexual activities). This survey of 434 men reported that lower overall sexual satisfaction and lower erectile function were associated with problematic Internet pornography use [44]. In addition, 20.3% of the men said that one motive for their pornography use was “to maintain arousal with my partner” [44]. In a finding that may indicate escalation of pornography use, 49% described sometimes “searching for sexual content or being involved in OSAs that were not previously interesting to them or that they considered disgusting” [44] (p.260). Finally, a significant percentage of the participants (27.6%) self-assessed their consumption of OSAs as problematic. While this rate of problematic pornography use may appear to be high, another 2016 study on 1298 men who had viewed pornography in the last six months reported that 28% of participants scored at or above the cutoff for hypersexuality disorder [49].

Our review also included two 2015 papers claiming that Internet pornography use is unrelated to rising sexual difficulties in young men. However, such claims appear to be premature on closer examination of these papers and related formal criticism. The first paper contains useful insights about the potential role of sexual conditioning in youthful ED [50]. However, this publication has come under criticism for various discrepancies, omissions and methodological flaws. For example, it provides no statistical results for the erectile function outcome measure in relation to Internet pornography use. Further, as a research physician pointed out in a formal critique of the paper, the papers’ authors, “have not provided the reader with sufficient information about the population studied or the statistical analyses to justify their conclusion” [51]. Additionally, the researchers investigated only hours of Internet pornography use in the last month. Yet studies on Internet pornography addiction have found that the variable of hours of Internet pornography use alone is widely unrelated to “problems in daily life”, scores on the SAST-R (Sexual Addiction Screening Test), and scores on the IATsex (an instrument that assesses addiction to online sexual activity) [52,53,54,55,56]. A better predictor is subjective sexual arousal ratings while watching Internet pornography (cue reactivity), an established correlate of addictive behavior in all addictions [52,53,54]. There is also increasing evidence that the amount of time spent on Internet video-gaming does not predict addictive behavior. “Addiction can only be assessed properly if motives, consequences and contextual characteristics of the behavior are also part of the assessment” [57]. Three other research teams, using various criteria for “hypersexuality” (other than hours of use), have strongly correlated it with sexual difficulties [15,30,31]. Taken together, this research suggests that rather than simply “hours of use”, multiple variables are highly relevant in assessment of pornography addiction/hypersexuality, and likely also highly relevant in assessing pornography-related sexual dysfunctions.

A second paper reported little correlation between frequency of Internet pornography use in the last year and ED rates in sexually active men from Norway, Portugal and Croatia [6]. These authors, unlike those of the previous paper, acknowledge the high prevalence of ED in men 40 and under, and indeed found ED and low sexual desire rates as high as 31% and 37%, respectively. In contrast, pre-streaming Internet pornography research done in 2004 by one of the paper’s authors reported ED rates of only 5.8% in men 35–39 [58]. Yet, based on a statistical comparison, the authors conclude that Internet pornography use does not seem to be a significant risk factor for youthful ED. That seems overly definitive, given that the Portuguese men they surveyed reported the lowest rates of sexual dysfunction compared with Norwegians and Croatians, and only 40% of Portuguese reported using Internet pornography “from several times a week to daily”, as compared with the Norwegians, 57%, and Croatians, 59%. This paper has been formally criticized for failing to employ comprehensive models able to encompass both direct and indirect relationships between variables known or hypothesized to be at work [59]. Incidentally, in a related paper on problematic low sexual desire involving many of the same survey participants from Portugal, Croatia and Norway, the men were asked which of numerous factors they believed contributed to their problematic lack of sexual interest. Among other factors, approximately 11%–22% chose “I use too much pornography” and 16%–26% chose “I masturbate too often” [60].

Again, intervention studies would be the most instructive. However, with respect to correlation studies, it is likely that a complex set of variables needs to be investigated in order to elucidate the risk factors at work in unprecedented youthful sexual difficulties. First, it may be that low sexual desire, difficulty orgasming with a partner and erectile problems are part of the same spectrum of Internet pornography-related effects, and that all of these difficulties should be combined when investigating potentially illuminating correlations with Internet pornography use.

Second, although it is unclear exactly which combination of factors may best account for such difficulties, promising variables to investigate in combination with frequency of Internet pornography use might include (1) years of pornography-assisted versus pornography-free masturbation; (2) ratio of ejaculations with a partner to ejaculations with Internet pornography; (3) the presence of Internet pornography addiction/hypersexuality; (4) the number of years of streaming Internet pornography use; (5) at what age regular use of Internet pornography began and whether it began prior to puberty; (6) trend of increasing Internet pornography use; (7) escalation to more extreme genres of Internet pornography, and so forth.

## 2. Clinical Reports

While correlation studies are easier to conduct, the difficulty in isolating the precise variables at work in the unprecedented rise of sexual dysfunction in men under 40 suggests that intervention studies (in which subjects removed the variable of Internet pornography use) would better establish whether there is a connection between its use and sexual difficulties. The following clinical reports demonstrate how asking patients with diverse and otherwise unexplained dysfunctions to eradicate Internet pornography use helps to isolate its effects on sexual difficulties. Below we report on three active duty servicemen. Two saw a physician for their non-organic erectile dysfunction, low sexual desire, and unexplained difficulty in achieving orgasm with partners. The first mentioned variables (1), (6) and (7), listed in the preceding paragraph. The second mentioned (6) and (7). Both were free of mental health diagnoses. We also report a third active duty serviceman who saw a physician for mental health reasons. He mentioned variable (6).

### 2.1. First Clinical Report

A 20-year old active duty enlisted Caucasian serviceman presented with difficulties achieving orgasm during intercourse for the previous six months. It first happened while he was deployed overseas. He was masturbating for about an hour without an orgasm, and his penis went flaccid. His difficulties maintaining erection and achieving orgasm continued throughout his deployment. Since his return, he had not been able to ejaculate during intercourse with his fiancée. He could achieve an erection but could not orgasm, and after 10–15 min he would lose his erection, which was not the case prior to his having ED issues. This was causing problems in his relationship with his fiancée.

Patient endorsed masturbating frequently for “years”, and once or twice almost daily for the past couple of years. He endorsed viewing Internet pornography for stimulation. Since he gained access to high-speed Internet, he relied solely on Internet pornography. Initially, “soft porn”, where the content does not necessarily involve actual intercourse, “did the trick”. However, gradually he needed more graphic or fetish material to orgasm. He reported opening multiple videos simultaneously and watching the most stimulating parts. When preparing for deployment about a year ago, he was worried about being away from partnered sex. So, he purchased a sex toy, which he described as a “fake vagina”. This device was initially so stimulating that he reached orgasm within minutes. However, as was the case with Internet pornography, with increased use, he needed longer and longer to ejaculate, and eventually he was unable to orgasm at all. Since returning from deployment, he reported continued masturbation one or more times per day using both Internet pornography and toy. Although physically and emotionally attracted to his fiancée, the patient reported that he preferred the device to actual intercourse because he found it more stimulating. He denied any other relationship issues. He also denied any personal and/or vocational stressors. He described his mood as “concerned” because he was worried there was something wrong with his genitals and he wanted his relationship with his fiancée to work. She was starting to think that he was no longer attracted to her.

Medically, he had no history of major illness, surgery, or mental health diagnoses. He was not taking any medications or supplements. He denied using tobacco products but drank a few drinks at parties once or twice a month. He had never blacked out from alcohol intoxication. He reported multiple sexual partners in the past, but since his engagement a year ago his fiancée had been his sole sexual partner. He denied a history of sexually transmitted diseases. On physical examination, his vital signs were all normal, and his genital exam was normal appearing without lesions or masses.

At the conclusion of the visit, it was explained to him that use of a sex toy had potentially desensitized his penile nerves and watching hardcore Internet pornography had altered his threshold for sexual stimulation. He was advised to stop using the toy and watching hardcore Internet pornography. He was referred to urology for further evaluation. By the time he was seen by the urologist a few weeks later, he had cut down on Internet pornography use significantly, although he said he could not completely stop. He ceased using the toy. He was having orgasms again through intercourse with his fiancée, and their relationship had improved. The urologist’s evaluation was normal.

### 2.2. Second Clinical Report

A 40-year old African American enlisted serviceman with 17 years of continuous active duty presented with difficulty achieving erections for the previous three months. He reported that when he attempted to have sexual intercourse with his wife, he had difficulty achieving an erection and difficulty maintaining it long enough to orgasm. Ever since their youngest child left for college, six months earlier, he had found himself masturbating more often due to increased privacy. He formerly masturbated every other week on average, but that increased to two to three times per week. He had always used Internet pornography, but the more often he used it, the longer it took to orgasm with his usual material. This led to him using more graphic material. Soon thereafter, sex with his wife was “not as stimulating” as before and at times he found his wife “not as attractive”. He denied ever having these issues earlier in the seven years of their marriage. He was having marital issues because his wife suspected he was having an affair, which he adamantly denied.

His medical history was only significant for hypertension, which was diagnosed more than two years earlier and had been well controlled with a diuretic: 25mg of chlorthalidone daily. He took no other medications or supplements. His only surgery was an appendectomy performed three years prior. He had no sexually transmitted diseases or mental health diagnoses. He endorsed smoking three packs of cigarettes per week for over ten years and drinking one to two drinks per week. Physical exam revealed vital signs within normal ranges, normal cardiovascular exam, and normal appearing genitals without lesions or masses.

At the end of the exam, his issues were attributed to heightened sexual stimulation threshold from exposure to hardcore Internet pornography and frequent masturbation. He was advised to stop watching hardcore Internet pornography and decrease masturbation frequency. Three months later, the patient reported that he tried “really hard” to avoid hardcore Internet pornography and to masturbate less, but he “just couldn’t do it”. He said whenever he was home alone, he found himself watching Internet pornography, which would eventually lead to masturbation. Not watching made him feel like he was “missing out”, which made him irritable and made him want to do it even more, to the point where he looked forward to his wife leaving the house. He was offered a referral to sex behavioral therapy, but he declined. He wanted to try to work on his behavior on his own.

### 2.3. Third Clinical Report

A 24-year old junior Enlisted Sailor was admitted to the inpatient mental health unit after a suicide attempt by overdose. During his evaluation and treatment he admitted to drinking alcohol even though he was advised to not use alcohol while being treated with antidepressant medications. His history and increasing tolerance were consistent with mild Alcohol Use Disorder due to his use while taking antidepressants. As part of the addictions portion of his history he was asked about gambling, Internet gaming and pornography addiction. He revealed that he had become concerned over his use of pornography, spending an excessive amount of time (5+ h a day) viewing online pornography for about six months. He also realized that he had diminished sexual interest in his wife, manifested by his inability to maintain sustained erections, preferring to view pornography where he had no erectile issues. When he became aware of his excessive use of pornography, he stopped viewing it completely, telling his interviewer he was afraid that if he viewed it to any extent he would find himself overusing it again. He reported that after he ceased using pornography his erectile dysfunction disappeared.

In summary, intervention studies designed to reveal causation by removing the variable of Internet pornography use are much needed to investigate unexplained sexual difficulties in Internet pornography users under 40. As suggested by our clinical reports, as well as the successes of clinicians Doidge [33] and Bronner and Ben-Zion [34] above, such research might ask study participants with possible PIED, difficulty achieving orgasm with a partner, and/or low sexual desire/satisfaction to eliminate Internet pornography.

## 3. Discussion

### 3.1. Male Sexual Response in the Brain

While male sexual response is complex, several key brain regions are critical for achieving and maintaining erections [61]. Hypothalamic nuclei play an important role in regulating sexual behavior and erections by acting as an integration center for brain and peripheral input [62]. The hypothalamic nuclei that facilitate erections receive pro-erectile input from the mesolimbic dopamine pathway, which comprises the ventral tegmental area (VTA) and the nucleus accumbens (NAc) [62]. The VTA-NAc circuit is a key detector of rewarding stimuli, and forms the core of a broader, more complex set of integrated circuits commonly called the “reward system” [63]. An individual’s response to natural rewards, such as sex, is largely regulated by the mesolimbic dopamine pathway, which receives excitatory and inhibitory input from other limbic structures and the prefrontal cortex [64]. Erections are dependent upon activation of dopaminergic neurons in VTA and dopamine receptors in the NAc [65,66]. Excitatory glutamate inputs from other limbic structures (amygdala, hippocampus) and the prefrontal cortex facilitate dopaminergic activity in the VTA and NAc [62]. Reward responsive dopamine neurons also project into the dorsal striatum, a region activated during sexual arousal and penile tumescence [67]. Dopamine agonists, such as apomorphine, have been shown to induce erection in men with both normal and impaired erectile function [68]. Thus, dopamine signaling in the reward system and hypothalamus plays a central role in sexual arousal, sexual motivation and penile erections [65,66,69].

We propose that chronic Internet pornography use resulted in erectile dysfunction and delayed ejaculation in our servicemen reported above. We hypothesize an etiology arising in part from Internet pornography-induced alterations in the circuits governing sexual desire and penile erections. Both hyper-reactivity to Internet pornography cues via glutamate inputs and downregulation of the reward system’s response to normal rewards may be involved. These two brain changes are consistent with chronic overconsumption of both natural rewards and drugs of abuse, and are mediated by dopamine surges in the reward system [70,71,72].

### 3.2. Internet Pornography as Supernormal Stimulus

Arguably, the most important development in the field of problematic sexual behavior is the way in which the Internet is influencing and facilitating compulsive sexual behavior [73]. Unlimited high-definition sexual videos streaming via “tube sites” are now free and widely accessible, 24 h a day via computers, tablets and smartphones, and it has been suggested that Internet pornography constitutes a supernormal stimulus, an exaggerated imitation of something our brains evolved to pursue because of its evolutionary salience [74,75]. Sexually explicit material has been around for a long time, but (1) video pornography is significantly more sexually arousing than other forms of pornography [76,77] or fantasy [78]; (2) novel sexual visuals have been shown to trigger greater arousal, faster ejaculation, and more semen and erection activity compared with familiar material, perhaps because attention to potential novel mates and arousal served reproductive fitness [75,79,80,81,82,83,84]; and (3) the ability to self-select material with ease makes Internet pornography more arousing than pre-selected collections [79]. A pornography user can maintain or heighten sexual arousal by instantly clicking to a novel scene, new video or never encountered genre. A 2015 study assessing Internet pornography’s effects on delay discounting (choosing immediate gratification over delayed rewards of greater value) states, “The constant novelty and primacy of sexual stimuli as particularly strong natural rewards make internet pornography a unique activator of the brain’s reward system. ... It is therefore important to treat pornography as a unique stimulus in reward, impulsivity, and addiction studies” [75] (pp. 1, 10).

Novelty registers as salient, enhances reward value, and has lasting effects on motivation, learning and memory [85]. Like sexual motivation and the rewarding properties of sexual interaction, novelty is compelling because it triggers bursts of dopamine in regions of the brain strongly associated with reward and goal-directed behavior [66]. While compulsive Internet pornography users show stronger preference for novel sexual images than healthy controls, their dACC (dorsal anterior cingulate cortex) also shows more rapid habituation to images than healthy controls [86], fueling the search for more novel sexual images. As co-author Voon explained about her team’s 2015 study on novelty and habituation in compulsive Internet pornography users, “The seemingly endless supply of novel sexual images available online [can feed an] addiction, making it more and more difficult to escape” [87]. Mesolimbic dopamine activity can also be enhanced by additional properties often associated with Internet pornography use such as, violation of expectations, anticipation of reward, and the act of seeking/surfing (as for Internet pornography) [88,89,90,91,92,93]. Anxiety, which has been shown to increase sexual arousal [89,94], may also accompany Internet pornography use. In short, Internet pornography offers all of these qualities, which register as salient, stimulate dopamine bursts, and enhance sexual arousal.

### 3.3. Internet Pornography Use as Self-Reinforcing Activity

As the reward system encourages organisms to remember and repeat critical behaviors, such as sex, eating, and socializing, chronic Internet pornography use may become a self-reinforcing activity [95]. The reward system is vulnerable to pathological learning [96], particularly in adolescents, such as greater risk of addiction [97,98] and greater future use of “deviant pornography” (bestiality and child pornography) [99]. Several lines of research have begun to elucidate the overlap in the neural substrates of sexual learning and addiction [100,101]. For example, sexual behaviors and addictive drugs activate the same sets of neurons within the same reward system structures (NAc, basolateral amygdala, anterior cingulated area) [102]. In contrast, very little overlap exists between other natural rewards (food, water) and addictive drugs, such as cocaine and methamphetamine [102]. Thus, methamphetamine use recruits the same mechanisms and neural substrates as does the natural reward of sexual stimulation [103]. In another study, cocaine addicts had nearly identical brain activation patterns when viewing pornography and cues related to their addiction, but brain activation patterns when viewing nature scenes were completely different [104].

Furthermore, both repeated sexual behaviors and repeated psychostimulant administration induce up regulation of Delta FosB, a transcription factor that promotes several neuroplastic changes that sensitize the mesolimbic dopamine system to the activity in question [103]. In both addictive drug use and sexual reward, this up regulation in the same NAc neurons is mediated via dopamine receptors [103]. This process renders the individual hyper-sensitized to stimuli associated with the activity (increased incentive salience) [105]. Exposure to related cues then triggers cravings to engage in the behavior (increased “wanting”), and may lead to compulsive use [106]. In comparing sexual reward to substances of abuse, researchers Pitchers et al. concluded that, “Natural and drug rewards not only converge on the same neural pathway, they converge on the same molecular mediators, and likely in the same neurons in the NAc, to influence the incentive salience and the “wanting” of both types of rewards” [103]. In the same vein, a 2016 review by Kraus, Voon and Potenza affirmed that, “Common neurotransmitter systems may contribute to [compulsive sexual behavior] and substance use disorders, and recent neuroimaging studies highlight similarities relating to craving and attentional biases” [107].

To date, the potential health risks of Internet pornography are not as well understood as those for alcohol and tobacco use, and Internet pornography use is widely portrayed as both ordinary behavior and increasingly socially acceptable [108,109]. Perhaps this is why men are slow to connect their pornography viewing with their sexual difficulties. After all, “Who doesn’t watch porn these days?” as one of our servicemen asked his physician. He regarded his problematic progression as normal, perhaps even evidence of high libido [110]. However, there is growing evidence that it was an indication of addiction-related processes [31,52,54,73,86,107,111,112,113,114,115,116,117,118,119,120,121,122]. Finnish researchers found “adult entertainment” to be the most common reason for compulsive Internet use [123], and a one-year longitudinal study of Internet applications revealed that Internet pornography may have the highest potential for addiction [124], with Internet gaming a close second in both studies. To date, Internet gaming disorder (IGD) has been slated for further study in the *Diagnostic and Statistical Manual of Mental Disorders* (DSM-5) [125], while Internet pornography addiction disorder has not. However, in the view of UK researcher Griffiths, “the empirical base for sex addiction is arguably on a par with IGD” [73]. In fact, various addiction experts are calling for Internet addiction to be recognized as a generalized problem with more specific subtypes such as gaming and pornography [118,126,127,128]. A 2015 review also concluded that Internet pornography addiction should be recognized as a subtype of Internet addiction, which belongs in the DSM [118].

Interestingly, our second serviceman meets many of the criteria proposed for IGD in the DSM-5, adjusted for Internet pornography use. He exhibited the following: (1) preoccupation with Internet pornography; (2) loss of interest in sex with his real-life partner as a consequence; (3) withdrawal symptoms such as irritability and resentment; (4) seeking pornography to relieve his bad feelings; (5) inability to quit despite severe problems; and (6) escalation to more graphic material.

### 3.4. Neuroadaptations Related to Internet Pornography-Induced Sexual Difficulties

We hypothesize that pornography-induced sexual difficulties involve both hyperactivity and hypoactivity in the brain’s motivational system [72,129] and neural correlates of each, or both, have been identified in recent studies on Internet pornography users [31,48,52,53,54,86,113,114,115,120,121,130,131,132,133,134]. We have broken this portion of our discussion into three somewhat interrelated sections.

#### 3.4.1. Increased Incentive Salience for Internet Pornography (Hyperactivity)

Hyperactivity refers to a sensitized, conditioned response to cues associated with use. Sensitized learning involves an enhanced mesolimbic dopamine system response that results in attribution of potentially pathological levels of incentive salience to cue-evoked seeking of drugs and natural rewards [135,136,137]. The mesolimbic dopamine system receives glutamate inputs from various cortical and limbic regions. Current theory suggests glutamatergic synapses associated with seeking and obtaining a particular reward undergo modifications, which enhance the response of the mesolimbic dopamine system to that same reward [100,138]. These powerful new learned associations underlie the “incentive-salience” (or “incentive motivation”) theory of addiction.

With respect to our servicemen’s contact with partners, it is possible that as they sensitized their sexual arousal to Internet pornography, partnered sex no longer met their conditioned expectations and no longer triggered the release of sufficient dopamine to produce and sustain erections [50,62,139]. As Prause and Pfaus note, “Erectile problems may occur when real-life sexual stimulation does not match the broad content [accessible online]” [50]. Human and animal studies suggest that when expectations are unmet (a negative prediction error), activity in the mesolimbic dopamine pathway is inhibited [140,141,142,143]. Addiction studies have reported that cues explicitly paired with the absence of drug reward can have marked inhibitory effects on dopamine release [72]. Consistent with a negative prediction error, Banca et al. reported a decrease in ventral striatal activity in response to the omission of an expected sexual image (following a conditioned cue) [86]. Banca et al. also reported that, compared to healthy controls, compulsive Internet pornography users had enhanced preference for conditioned cues (abstract patterns) related to sexual images [86]. This finding suggests that Internet pornography users can become sensitized to cues that are unrelated to sexual content, associations that can be extremely challenging to extinguish [87].

A 2014 fMRI study by Voon et al. provides support for the incentive-salience (sensitization) model with respect to compulsive Internet pornography users [31]. Compared to healthy controls, compulsive Internet pornography users had enhanced activity to sexually explicit films in the ventral striatum, amygdala and dorsal anterior cingulate cortex. This same core network is activated during cue reactivity and drug craving in substance abusers [144]. Voon et al. also reported that, “Compared to healthy volunteers, [compulsive internet pornography users] had greater subjective sexual desire or wanting to explicit cues and had greater liking scores to erotic [less explicit] cues, thus demonstrating a dissociation between wanting and liking” [31] (p. 2). In the incentive-sensitization model of addiction, dissociation between “wanting” and “liking” is considered an indication of pathological learning [106]. As the addiction to explicit Internet pornography progresses, motivation and cravings to use (“wanting”) increase, while pleasure from its use (“liking”) decreases. Here, Internet pornography viewers “liked” the tamer erotic stimuli, but “wanted” the explicit cues disproportionately. Similar to our servicemen, the majority of Voon et al.’s subjects (mean age 25) “had greater impairments of sexual arousal and erectile difficulties in intimate relationships but not with sexually explicit materials highlighting that the enhanced desire scores were specific to the explicit cues and not generalized heightened sexual desire” [31] (p. 5). A related study on most of the same subjects found enhanced attentional bias in compulsive Internet pornography users similar to that observed in studies of drug cues in addiction disorders [111]. The research team concluded that, “These studies together provide support for an incentive motivation theory of addiction underlying the aberrant response towards sexual cues in CSB [compulsive sexual behavior]” [111].

A 2015 fMRI study on male hypersexuals by Seok and Sohn replicated and expanded upon the findings of Voon et al. [31] and Mechelmans et al. [111] , just described [120]. Seok and Sohn reported that compared to controls hypersexuals had significantly greater brain activation when exposed to sexual images for 5 s. While Voon et al [31] examined cue-induced activity in the dACC-ventral striatal-amygdala functional network, Seok and Sohn assessed activity in the dorsolateral prefrontal cortex (DLPFC), caudate nucleus, inferior parietal lobe, dorsal anterior cingulate gyrus, and the thalamus. Seok and Sohn added that the severity of sexual addiction directly correlated with cue-induced activation of the DLPFC and thalamus. A third finding was that compared to controls hypersexuals had far greater DLPFC activation to sexual cues, yet far less DLPFC activation to neutral stimuli. This mirrors abnormal prefrontal cortex functioning in individuals with addiction where increased sensitivity to addiction cues is coupled with less interest in normal rewarding activities [145]. This finding aligns with our hypothesis that both hyperactivity and hypoactivity of the brain’s motivational system are involved in compulsive pornography use, and may be related to pornography-induced sexual dysfunctions.

A 2016 fMRI cue-reactivity study on male heterosexual pornography users expanded on previous findings [54]. Brand et al. reported that ventral striatum activity was greater for preferred pornographic material as compared to non-preferred pornographic material. In addition, stronger ventral striatum activity for preferred pornographic material was related to self-reported symptoms of addictive use of Internet pornography. In fact, symptoms of Internet pornography addiction (as assessed by the s-IATsex) were the only significant predictor of ventral striatum response to preferred versus non-preferred pornographic pictures. Other variables, such as weekly amount of cybersex, sexual excitability, hypersexual behavior in general, symptoms of depression and interpersonal sensitivity, and indicators of intensity of current sexual behavior, did not relate to cue-induced ventral striatum activity. Put simply, it was sensitization that best predicted symptoms of Internet pornography addiction. Brand et al. concluded that, “The findings emphasize parallels between IPA [internet pornography addiction] and other behavioral addictions and substance-related disorders” [54].

A 2016 fMRI study (Klucken et al.) [121] compared two groups of heterosexual males: subjects with compulsive sexual behaviors (CSB) and healthy controls. The mean time typically spent watching sexually explicit material weekly was 1187 min for the CSB group and 29 min for the control group. Researchers exposed all subjects to a conditioning procedure in which previously neutral stimuli (colored squares) predicted the presentation of an erotic picture. Compared to controls the subjects with CSB displayed increased activation of the amygdala during presentation of the conditioned cue predicting the erotic picture. This finding aligns with studies reporting increased amygdala activation when substance abusers are exposed to cues related to drug use [146]. Voon et al. also reported that explicit videos induced greater amygdala activation in CSB subjects than in healthy controls. This research converges with animal research linking the amygdala to appetitive conditioning. For example, stimulating opioid circuitry in the amygdala magnifies incentive salience intensity towards a conditioned cue, accompanied by a simultaneous reduction of the attractiveness of an alternative salient target [147]. While the CSB group in Klucken et al. [121] had greater amygdala activation to a cue predicting a sexual image, their subjective sexual arousal was no higher than controls. Interestingly, three of the twenty CSB subjects reported “orgasmic-erection disorder” when interviewed to screen for Axis I and Axis II diagnoses, while none of the control subjects reported sexual problems. This finding recalls Voon et al., in which CSB subjects had greater amygdala-ventral striatum-dACC activation to explicit sexual videos, yet 11 of 19 reported erectile or arousal difficulties with sexual partners. Klucken et al. also found decreased coupling between the ventral striatum and the prefrontal cortex in subjects with CSB compared with controls. Decreased ventral striatal-PFC coupling has been reported in substance disorders and is believed to be related to impaired impulse control [145].

A 2013 EEG study by Steele et al. reported higher P300 amplitude to sexual images, relative to neutral pictures, in individuals complaining of problems regulating their Internet pornography use [48]. Substance abusers also exhibit greater P300 amplitude when exposed to visual cues associated with their addiction [148]. In addition, Steele et al. reported a negative correlation between P300 amplitude and desire for sex with a partner [48]. Greater cue reactivity to Internet pornography paired with less sexual desire for partnered sex, as reported by Steele et al., aligns with the Voon et al. finding of “diminished libido or erectile function specifically in physical relationships with women” in compulsive Internet pornography users [31]. Supporting these findings, two studies assessing sexual desire and erectile function in “hypersexuals” and compulsive Internet pornography users reported associations between measures of hypersexuality, and reduced desire for partnered sex and sexual difficulties [15,30]. Additionally, the 2016 survey of 434 men who viewed Internet pornography at least once in the last three months reported that problematic use was associated with higher levels of arousabilty, yet lower sexual satisfaction and poorer erectile function [44]. These results should be viewed in light of the multiple neuropsychology studies that have found that sexual arousal to Internet pornography cues and cravings to view pornography were related to symptom severity of cybersex addiction and self-reported problems in daily life due to excessive Internet pornography use [52,53,54,113,115,149,150]. Taken together, multiple and varied studies on Internet pornography users align with the incentive-salience theory of addiction, in which changes in the attraction value of an incentive correspond with changes in activation of regions of the brain implicated in the sensitization process [31,106]. To sum up, in alignment with our hypothesis, various studies report that greater reactivity toward pornographic cues, cravings to view, and compulsive pornography use are associated with sexual difficulties and diminished sexual desire for partners.

#### 3.4.2. Decreased Reward Sensitivity (Hypoactivity)

In contrast with the hyperactive response to Internet pornography cues just described, hypoactivity is a concomitant decrease in reward sensitivity to normally salient stimuli [70,151,152,153], such as partnered sex [31,48]. This decrease is also behind tolerance [70], and has been implicated in both substance and behavioral addictions [153,154,155,156], including other types of Internet addictions [157,158,159].Our servicemen’s tolerance to Internet pornography increased fairly quickly, leading to viewing more extreme material. The fact that self-selected pornography video is more arousing than other pornography may contribute to habituation or tolerance [27,75,79,81,160]. For example, men who viewed a sexual film rather than a neutral film later showed less response to sexual images, a possible indication of habituation [161]. Not long after pornography videotapes became available, researchers also discovered that when viewers were given ad libitum access to pornography videotapes of varying themes they swiftly escalated to more extreme pornography [162]. The more video pornography viewed, the greater the desire for hardcore themes [27,43,162], indicative of declining sexual responsiveness. (Again, half of Kinsey Institute subjects who regularly consumed video pornography showed little erectile responsiveness in the lab, and reported a need for more novelty and variety [27], and half of pornography users surveyed recently also had moved to material that did not interest them previously or which they found disgusting [44] (p. 260).) In another study, sexual satisfaction with partners, as measured by affection, physical appearance, sexual curiosity, and sexual performance, was inversely related to pornography use [43]. In pair-bonding mammals extreme stimulation with amphetamine impairs pair-bonding via activation of mesolimbic dopamine receptors [163], and it is possible that today's supernormally stimulating Internet pornography brings about a similar effect in some users.

In line with the suggestion that some Internet pornography users’ reward systems may be hypoactive in response to partnered sex (as well as hyper-reactive to cues for Internet pornography use), a 2014 fMRI study of non-compulsive Internet pornography users by Kühn and Gallinat found that the right caudate of the striatum was smaller with more hours and years of Internet pornography viewing [134].The caudate appears to be involved in approach-attachment behaviors and is strongly implicated in motivational states associated with romantic love [164,165]. Also, the greater the subjects’ Internet pornography use, the lower the activation in the left putamen when viewing sexually explicit still photos (0.530 s exposure). Activation of the putamen is associated with sexual arousal and penile tumescence [67,166]. The authors suggested both findings were “in line with the hypothesis that intense exposure to pornographic stimuli results in a downregulation of the natural neural response to sexual stimuli” [134]. Interestingly, men with a “higher interest in degrading or extreme pornography” report greater concerns about their sexual performance, penis size, and ability to sustain an erection than other Internet pornography users [42]. As hypothesized, extreme pornography viewing may decrease sexual responsiveness in some users, thus driving a spiraling need for more extreme or novel material to perform [27]. Again, a 2016 study reported that half of men surveyed had moved to material “not previously interesting to them or that they considered disgusting” [44].

A 2015 EEG study by Prause et al. compared frequent viewers of Internet pornography (mean 3.8 h/week) who were distressed about their viewing to controls (mean 0.6 h/week) as they viewed sexual images (1.0 s exposure) [130]. In a finding that parallels Kühn and Gallinat, frequent Internet pornography viewers exhibited less neural activation (LPP) to sexual images than controls [130]. The results of both studies suggest that frequent viewers of Internet pornography require greater visual stimulation to evoke brain responses when compared with healthy controls or moderate Internet pornography users [167,168]. In addition, Kühn and Gallinat reported that higher Internet pornography use correlated with lower functional connectivity between the striatum and the prefrontal cortex. Dysfunction in this circuitry has been related to inappropriate behavioral choices regardless of potential negative outcome [169]. In line with Kühn and Gallinat, neuropsychological studies report that subjects with higher tendency towards cybersex addiction have reduced executive control function when confronted with pornographic material [53,114].

A 2015 fMRI study by Banca et al. reported that, compared to healthy controls, compulsive Internet pornography subjects had a greater choice preference for novel sexual images [86]. While novelty-seeking and sensation-seeking are associated with greater risk for several types of addictions [170], Banca et al. found no differences in sensation-seeking scores between compulsive Internet pornography users and healthy controls. The authors suggest that the preference for novelty was specific to Internet pornography use, and not generalized novelty- or sensation-seeking [86]. These results align with Brand et al. (2011), which found that “the number of sex applications used” was a significant predictor of addiction using the IATsex questionnaire, while personality facets were not related to cybersex addiction [53]. Banca et al. also reported that compulsive Internet pornography users showed greater habituation in the dorsal anterior cingulate cortex (dACC) to repeated viewing of the same sexual images [86]. Generally speaking, the degree of dACC habituation to sexual images was associated with greater preference for novel sexual stimuli [86]. The dACC is implicated in drug cue reactivity and craving, as well as the assessment of expected versus unexpected rewards [144,171]. Voon et al. reported enhanced dACC activity in compulsive Internet pornography subjects in response to sexually explicit videos [31]. Banca et al.’s findings strongly suggest that greater novelty seeking in compulsive Internet pornography users is driven by more rapid habituation to sexual stimuli. The researchers concluded, “We show experimentally what is observed clinically that [compulsive internet pornography use] is characterized by novelty-seeking, conditioning and habituation to sexual stimuli in males” [86]. In a related study, many of these same subjects had also reported sexual arousal and erectile difficulties in partnered sexual activity, but not during Internet pornography use [31]. This implies that Internet pornography-induced sexual difficulties may be partly due to conditioned expectations of novelty that are not matched in partnered sexual activity. Taken together, Kühn and Gallinat [134], Prause et al. [130] and Banca et al. [86] demonstrated that frequent Internet pornography users exhibit (1) less brain activation in response to brief exposure to sexual images; (2) greater preference for novel sexual stimuli; (3) faster dACC habituation to sexual stimuli; and (4) less grey matter volume in the caudate. These findings support the hypothesis that Internet pornography use may decrease reward sensitivity, leading to increased habituation and tolerance as well as the need for greater stimulation to become sexually aroused.

Studies investigating psychogenic ED provide further support for the role of reward system hypoactivity in erectile dysfunction and low libido. Dopamine agonist apomorphine elicits penile erections in men with psychogenic ED [172]. When a 2003 fMRI study monitored brain patterns while men with psychogenic ED and potent controls viewed sexual films, those with psychogenic ED differed significantly from potent controls in the degree of activation of cortical and subcortical regions. When dopamine agonist apomorphine was administered to men with psychogenic ED, it produced brain activation patterns similar to those seen in potent controls: significantly increased striatal and hypothalamic activity combined with cortical deactivation [173]. Moreover, a 2012 MRI study found a strong correlation between a reduction of striatal and hypothalamic grey matter and psychogenic ED [174]. A 2008 study reported men with psychogenic ED exhibited blunted hypothalamic activity in response to a sexual film [175].

#### 3.4.3. Internet Pornography and Sexual Conditioning

Given that our servicemen reported that they experienced erections and arousal with Internet pornography, but not without it, research is needed to rule out inadvertent sexual conditioning as a contributing factor to today’s rising rates of sexual performance problems and low sexual desire in men under 40. Prause and Pfaus have hypothesized that sexual arousal may become conditioned to aspects of Internet pornography use that do not readily transition to real-life partner situations. “It is conceivable that experiencing the majority of sexual arousal within the context of VSS [visual sexual stimuli] may result in a diminished erectile response during partnered sexual interactions...When high stimulation expectations are not met, partnered sexual stimulation is ineffective” [50]. Such inadvertent sexual conditioning is consistent with the incentive-salience model. Several lines of research implicate increased mesolimbic dopamine in sensitization to both drugs of abuse and sexual reward [100,103]. Acting through dopamine D1 receptors, both sexual experience and psychostimulant exposure induce many of the same long-lasting neuroplastic changes in the NAc critical for enhanced wanting of both rewards [103].

Today’s Internet pornography user can maintain high levels of sexual arousal, and concomitant elevated dopamine, for extended periods due to unlimited novel content. High dopamine states have been implicated in conditioning sexual behavior in unexpected ways in both animal models [176,177] and humans. In humans, when Parkinson’s patients were prescribed dopamine agonists, some reported uncharacteristic compulsive pornography use and demonstrated greater neural activity to sexual picture cues, correlating with enhanced sexual desire [178]. Two recent fMRI studies reported that subjects with compulsive sexual behaviors are more prone to establish conditioned associations between formally neutral cues and explicit sexual stimuli than controls [86,121]. With repeated Internet pornography exposure, “wanting” may increase for Internet pornography’s expected novelty and variety, elements difficult to sustain during partnered sex. In line with the hypothesis that Internet pornography use can condition sexual expectations, Seok and Sohn found that compared to controls hypersexuals had greater DLPFC activation to sexual cues, yet less DLPFC activation to non-sexual stimuli [120]. It also appears that Internet pornography use can condition the user to expect or “want” novelty. Banca et al. reported that subjects with compulsive sexual behaviors had greater preference for novel sexual images and showed greater habituation in the dorsal anterior cingulate cortex to repeated viewing of the same sexual images [86]. In some users, a preference for novelty arises from the need to overcome declining libido and erectile function, which may, in turn, lead to new conditioned pornographic tastes [27].

When a user has conditioned his sexual arousal to Internet pornography, sex with desired real partners may register as “not meeting expectations” (negative reward prediction) resulting in a corresponding decline in dopamine. Combined with the inability to click to more stimulation, this unmet prediction may reinforce an impression that partnered sex is less salient than Internet pornography use. Internet pornography also offers a voyeur's perspective generally not available throughout partnered sex. It is possible that if a susceptible Internet pornography user reinforces the association between arousal and watching other peoplehave sex on screens while he is highly aroused, his association between arousal and real-life partnered sexual encounters may weaken.

Research on conditioning of sexual response in humans is limited, but shows that sexual arousal is conditionable [179,180,181], and particularly prior to adulthood [182]. In men, arousal can be conditioned to particular films [183], as well as to images [184]. Sexual performance and attraction in male (non-human) animals can be conditioned to an array of stimuli that are not typically sexually salient for them, including fruit/nut scents, aversive scents, such as cadaverine, same-sex partners, and the wearing of rodent jackets [177,185,186,187]. For example, rats that had learned sex *with* a jacket did not perform normally *without* their jackets [187].

In line with these conditioning studies, the younger the age at which men first began regular use of Internet pornography, and the greater their preference for it over partnered sex, the less enjoyment they report from partnered sex, and the higher their current Internet pornography use [37]. Similarly, men reporting increased consumption of bareback anal pornography (in which actors do not wear condoms) and its consumption at an earlier age, engage in more unprotected anal sex themselves [188,189]. Early consumption of pornography may also be associated with conditioning tastes to more extreme stimulation [99,190].

A review by Pfaus points to early conditioning as critical for sexual arousal templates: “It is becoming increasingly clear that there is a critical period of sexual behavior development that forms around an individual’s first experiences with sexual arousal and desire, masturbation, orgasm, and sexual intercourse itself” [191] (p. 32). The suggestion of a critical developmental period is consistent with the report of Voon et al. that younger compulsive Internet pornography users showed greater activity in the ventral striatum in response to explicit videos [31]. The ventral striatum is the primary region involved in sensitization to natural and drug reward [103]. Voon et al. also reported that compulsive Internet pornography subjects first viewed Internet pornography much earlier (mean age 13.9) than healthy volunteers (mean age 17.2) [31]. A 2014 study found that nearly half of college-age men now report they were exposed to Internet pornography prior to age 13, as compared with only 14% in 2008 [37]. Could increased Internet pornography use during a critical developmental phase increase the risk of Internet pornography-related problems? Might it help explain the 2015 finding that 16% of young Italian men who used Internet pornography more than once a week reported low sexual desire, compared with 0% in non-consumers [29]? Our first serviceman was only 20 and had been using Internet pornography since he gained access to high-speed Internet.

Males can successfully condition their sexual response in the laboratory with instructional feedback, but without further reinforcement, such laboratory-induced conditioning disappears in later trials [176]. This inherent neuroplasticity may suggest how two of our servicemen restored attraction and sexual performance with partners after abandoning a sex toy and/or cutting back on Internet pornography. Decreasing or extinguishing conditioned responses to artificial stimuli potentially restored attraction and sexual performance with partners.

## 4. Conclusions and Recommendations

Traditional factors that once explained sexual difficulties in men appear insufficient to account for the sharp rise in sexual dysfunctions and low sexual desire in men under 40. Both the literature and our clinical reports underscore the need for extensive investigation of Internet pornography's potential effects on users, ideally by having subjects remove the variable of Internet pornography in order to demonstrate potential effects of behavioral modification. A 2015 study, for example, found that rates of delay discounting (choosing immediate gratification over delayed rewards of greater value) decreased when healthy participants endeavored to give up Internet pornography use for just three weeks (compared with a control group who endeavored to give up their favorite food for the same time period) [75]. Both behavior and the nature of the stimuli given up were key variables.

While non-organic sexual dysfunctions have been presumed psychological in origin, and therefore the province of mental health experts, the unexplained sexual dysfunctions now rising sharply in young men (ED, difficulty orgasming, low sexual desire) are, to the extent they are reversible by quitting Internet pornography, not arising from “performance anxiety” (that is, psychosexual dysfunction, ICD-9 code 302.7), although performance anxiety may certainly accompany them. Future researchers will need to take into account the unique properties and impact of today's streaming Internet delivery of pornography. In addition, Internet pornography consumption during early adolescence, or before, may be a key variable.

Our review and clinical reports also highlight the need for validated screening tools to identify the possible presence of non-organic sexual difficulties, as well as Internet pornography-related difficulties in otherwise healthy men. The latter may often be reversible simply by modifying behavior. Because Internet pornography-related sexual difficulties are not yet specifically encompassed in an official diagnosis, healthcare providers do not routinely screen for them, leaving patients vulnerable. In this regard, in order to assess patients correctly, it may be critical to distinguish pornography-free from pornography-assisted masturbation. Traditionally, if patients had no difficulty with erections, arousal and climax while masturbating, but reported problems during partnered sex, they were presumed to have psychogenic, not organic, problems. However, young patients asked about their capabilities may assume “masturbation” refers to “masturbation with the aid of internet pornography”, and therefore be assessed as having “performance anxiety”, when their partnered-sex difficulties are actually Internet pornography-related. One simple test healthcare providers might employ is to ask, “whether the patient can achieve and sustain a satisfactory erection (and climax as desired) when masturbating without using Internet pornography”. If he cannot, but can easily achieve these goals with Internet pornography, then his sexual dysfunction may be associated with its use. Without employing such a test, there is a risk of false diagnoses of “performance anxiety”, and a consequent risk of prescribing needless psychoactive medications and (ultimately perhaps ineffective) phosphodiesterase-5 inhibitors. Other indications of Internet pornography-related performance difficulties may be loss of nocturnal erections and/or spontaneous erections. Additional research in this area is warranted.

Additionally, while healthcare providers must certainly screen for relationship problems, low self-esteem, depression, anxiety, PTSD, stress and other mental health problems, they should be cautious of assuming that poor mental health is the cause of otherwise unexplained sexual dysfunction in men under 40. The relationship between these factors and sexual dysfunction in young men may be bidirectional and co-occurring, or may be the consequence of sexual dysfunction [26].

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
