# Peer review of "Is Internet Pornography Causing Sexual Dysfunctions? A Review with Clinical Reports"

_behavsci, 2016, doi:10.3390/bs6030017_

Round 1

Reviewer 1 Report

Using three case studies as examples, the authors’ present a comprehensive literature review on the current research related to pornography induced erectile dysfunction (PIED).  Correctly asserting that ‘tube sites’ dramatically altered the content, delivery and intensity of online pornography beginning in 2006, the authors argue that internet pornography needs to be considered as a possible factor in the startlingly rise of ED among men under 40.  The review of the literature on ED and pornography is comprehensive and well argued, if a bit long and slightly repetitive.  However, the level of detail adds weight to the convincing case made in the article for the inclusion of online pornography in ED diagnoses.  My primary concern with the article has to do with the three case studies.  The case studies seem irrelevant in that they do not answer the methodological problems with other studies outlined in the beginning of the article nor do they meaningfully elaborate on the neurological connections outlined in the discussion.  This article seems more like a comprehensive literature review rather than a research article.  My suggestion is to alter the stated intent of the article to better fit its strengths.  For example, the abstract implies that the article is primarily a research article on three case studies when, in fact, the case studies play a minor role.  The authors can assert the hypotheses based on current literature and call for more research.  The case studies could be used as examples for how the hypothesis may be seen in patients as a way of integrating the case into the central argument. 

Overall, the piece is very strong on literature and makes a convincing case for its main argument about the relevancy of internet pornography to the health of young men’s sexual functioning.  But, the case studies, as currently framed, seem distracting and irrelevant.  With a bit of repositioning however, they can make a meaningful contribution.

Author Response

Thank you for your recommendations. Please see attached letter.

Andrew Doan, MD, PhD

Author Response

Thank you for your recommendations. Please see attached letter.

We have added additional references to support discussion about low sexual desire and also listed performance anxiety as a cause for psychogenic ED.

Thank you,

Andrew Doan, MD, PhD
